# Development of Water-Compatible Molecularly Imprinted Polymers Based on Functionalized β-Cyclodextrin for Controlled Release of Atropine

**DOI:** 10.3390/polym12010130

**Published:** 2020-01-06

**Authors:** Yahui He, Shaomei Zeng, A. M. Abd El-Aty, Ahmet Hacımüftüoğlu, Woldemariam Kalekristos Yohannes, Majid Khan, Yongxin She

**Affiliations:** 1China-Canada Joint Lab of Food Nutrition and Health (Beijing), School of Food and Health, Beijing Technology &Business University, Beijing 100048, China; 1770201321@st.btbu.edu.cn (W.K.Y.); majidkhanfst49@gmail.com (M.K.); 2Institute of Quality Standards & Testing Technology for Agro-Products, Chinese Academy of Agricultural Sciences, Beijing 100081, China; zengshaomei01@163.com; 3State Key Laboratory of Biobased Material and Green Papermaking, College of Food Science and Engineering, Qilu University of Technology, Shandong Academy of Science, Jinan 250353, China; 4Department of Pharmacology, Faculty of Veterinary Medicine, Cairo University, Giza 12211, Egypt; 5Department of Medical Pharmacology, Medical Faculty, Ataturk University, 25240 Erzurum, Turkey; ahmeth@atauni.edu.tr

**Keywords:** molecularly imprinted polymers, β-cyclodextrin, water-compatible, Atropine, controlled release

## Abstract

Herein, a novel method for molecularly imprinted polymers (MIPs) using methacrylic acid functionalized beta-cyclodextrin (MAA-β-CD) monomer is presented, which was designed as a potential water-compatible composite for the controlled release of atropine (ATP). The molecularly imprinted microspheres with pH-sensitive characteristics were fabricated using thermally-initiated precipitation polymerization, employing ATP as a template molecule. The effects of different compounds and concentrations of cross-linking agents were systematically investigated. Uniform microspheres were obtained when the ratio between ATP, MAA-β-CD, and trimethylolpropane trimethacrylate (TRIM) was 1:4:20 (mol/mol/mol) in polymerization system. The ATP loading equilibrium data was best suited to the Freundlich and Langmuir isotherm models. The *in vitro* drug release study was assessed under simulated oral administration conditions (pH 1.5 and 7.4). The potential usefulness of MIPs as drug delivery devices are much better than non-molecularly imprinted polymers (NIPs). The study shows that the prepared polymers are a pH stimuli-responsive system, which controlled the release of ATP, indicating the potential applications in the field of drug delivery.

## 1. Introduction

Atropine (ATP) is a naturally occurring alkaloid produced by plants from Solanaceae family [1]. ATP is classified as anticholinergic (parasympatholytic) agent and was originally used to dilate the pupils during cataract surgery; it also acted as a physiological antidote in acute poisoning caused by morphine (opium), eserine (physostigmine), and muscarine (mushroom poisoning) [2,3,4,5,6,7,8]. Multiple daily dosing regimens are required to maintain a steady-state concentration of a given remedy in plasma. Traditional dosage forms may result in treatment that may last for a short time or a longer period of zero-time drug concentration, when a dose is missed. Therefore, the design of a controlled release formulation is a major challenge to control the sustained release and delivery of medicine throughout the body. Therefore, there has been continued technological efforts and great motivation to improve the overall therapeutic effectiveness of sustained release drug delivery devices for atropine [9]. 

To maximize the safety and efficacy of a medication, it is worth considering the development of drug delivery system (DDS) that would regulate the rate of drug release to a specific site in the body [10]. Recently, the use of green polymeric materials in DDS has gained considerable attention. In this context, molecular imprinting is a well-known technique for creating polymeric materials with high recognition ability for template molecule as well as its structural analogues [11]. Molecularly imprinted polymers (MIPs), prepared in organic solvents with high proportion of crosslinkers, are considered a basic excipient for controlled release devices used in medicines with narrow therapeutic indices. For extended release with selective affinity to various biomolecules (such as proteins, peptides, and nucleic acids), MIPs are most widely used in biosensors [12,13,14,15], catalysis [16], antibody mimics [17] and molecular recognition [18], drug delivery [19,20], bio-separation [21], as well as diagnostics and therapeutic applications [22]. Among these applications, MIPs have attracted much attention in DDS for their binding tendency, specific recognition, and flexibility [13,23]. MIP can be directly used in drug delivery devices, as the particles can be suspended in aqueous solutions or membranes and used as a material in anti-bacterial coatings, and biosensors [24,25,26]. 

There are many reports on constructing a MIPs system using antibiotics as template, enhancing the loading capacity of target molecule and facilitating the sustained release of medicines [27,28]. Unfortunately, the competition between water molecules and the template creates weaker or destroying non-covalent bonds between template and functional monomer. This, in turn, makes the traditional MIPs exhibit poor site accessibility and low binding capacity for a template in water [29]. To solve these pitfalls, β-cyclodextrin (β-CD) can be used to modify the functional monomer; making the MIP have a greater binding affinity in an aqueous medium [30,31]. β-cyclodextrin is a cyclic oligosaccharide composed of seven D-glucose units linked by α-(1,4) glycosidic bonds. It is amphiphilic with a hydrophilic external, consisting of the hydrophobic molecular cavity surrounded by two hoops of hydroxyl groups. Due to its hydrophobic cavity and moderate molecular dimension, inclusion complexes are selectively formed with some molecules by β-CD through interaction between host and guest in water [32,33]. β-CD MIPs were successfully produced by bridging methacrylic acid (MAA) with β-CD [34]. Through the linkage between MAA monomer and β-CD, possible bonds between the template and monomers can be created [35,36], which result in improvement of selective adsorption and recognition affinity to MIPs. Thus, it is a promising step to prepare water-compatible molecularly imprinted polymers via modifying the monomer with β-CD.

To date, there are no published studies on the development of biodegradable MIP microspheres for in vitro controlled release of ATP. Furthermore, water-compatible molecularly imprinted polymers remain a challenge because of relatively weak interactions between electrostatic and hydrogen bonds in polar media, in addition to low binding affinity and selectivity to target analyte [37]. Stimuli-responsive MIPs not only have the ability to respond to external conditions (such as temperature, pH, magnetic field interference, and others), but they also have molecular recognition ability for targets molecules [38,39,40]. Therefore, the pH-sensitive MIP nanospheres as DDS has gained wide attention owing to selectivity, high loading capability, and effective controlled-release ability. In this study, methacrylic acid functionalized β-cyclodextrin was synthesized; afterwards, we developed β-cyclodextrin nano-sized MIP to control the release of atropine. The ATP loading capacity and the in vitro evaluation of pH-responsive controlled release system were systematically investigated. The principle of functionalized β-cyclodextrin in MIP to control the release of atropine is shown in Figure 1. 

## 2. Material and Methods

### 2.1. Chemicals and Materials

Certified standards of atropine sulfate, rutin, and anisodine hydrobromide were obtained from the National Institutes for Food and Drug Control (Beijing, China). β-Cyclodextrin (β-CD) was supplied by MP Biomedical (Illkirch Cedex, France). Trimethylolpropane trimethacrylate (TRIM), α,α′-azobisisobutyronitrile (AIBN), methacrylic acid (MAA), 2,4-Toluene diisocyanate (TDI), Dibutyltin dilaurate (DBTDL), and genipin were procured from Sigma Aldrich (St. Louis, MO, USA). HPLC grade formic acid, methanol, and acetonitrile were acquired from Merck (Darmstadt, Germany). Ultrapure water was obtained from Milli-Q water purification system (Millipore, Bedford, MA, USA). Dimethylacetamide (DMAC), epichlorohydrin (EPI), and other agents (analytical grade) were provided by Chemical Reagent Company (Beijing, China). A standard solution of ATP (1000 mg L^−1^) was prepared in methanol and stored at 4 °C in the dark.

### 2.2. Liquid Chromatography-Tandem Mass Spectrometry (LC-MS/MS) Conditions

The concentration of ATP was quantified by liquid chromatography-tandem mass spectrometry (LC-MS/MS). The chromatographic system was consisted of Agilent 1200 series coupled with API 2000 triple quadrupole mass spectrometer (AB SCIEX, Foster City, CA, USA), equipped with an electrospray ionization (ESI) interface. Chromatographic separation was performed on a reversed- phase C18 column (150 × 2.1 mm, 5 μm) at 30 °C with gradient elution using a mixture of mobile phase (A) 0.1% formic acid (*v/v*) in distilled water and (B) methanol. The gradient starting conditions were set at 90% A and 10% B; maintained for 3 min. Afterward, the percentage of mobile phase (B) was increased to 90% in 3 min to 7 min and ramped back to 10% at a flow rate of 200 μL min^−1^ and injection volume of 5 μL. Electrospray interface (ESI) was operated in positive ion mode, and the typical ESI parameters used were as follows: ion spray voltage (IS), 5500 V; atomization air pressure (GS1), 25 psi; auxiliary gas pressure (GS2), 40 psi; curtain gas (CUR), 40 psi; ion source temperature (TEM), 500 °C; entrance potential (EP), 10 V; and collision cell exit potential (CXP), 4 V. The MRM transitions, collision energy (CE), and declustering potential (DP) are summarized in Table 1. Calibration curves were attained by analyzing series of the standards solution for ATP and ASD (0.005–0.2 mg L^−1^) with LC-MS/MS. 

### 2.3. Synthesis of MAA-β-CD Monomer

The MAA-β-CD monomer was prepared according to the procedure reported earlier [41]. In brief, a psychometric ratio of 0.5 M MAA/1 M TDI/0.5 M β-CD, 2.855 mL of TDI, and 0.848 mL of MAA were mixed with 20 mL dimethylacetamide (DMAC) solvent, followed by the addition of DBTDL (0.1%, 0.02 mL) as a catalyst. After degassing with N_2_ for 10 min, the solution was magnetically stirred at room temperature for 1 h. Afterward, a further 5 mL of DMAC and 0.5 M of β-CD were added, and the mixture was kept under stirring for 2 h. The obtained product (MAA-β-CD) was purified by precipitation with methanol (20 mL) and subsequently re-precipitated in ultra-purified cold water. This process was repeated three times. 

### 2.4. Preparation Molecularly Imprinted Polymers (MIPs) and Non-Molecularly Imprinted Polymers (NIPs) Using Thermally Initiated Precipitation Polymerization 

The preparation of mono-dispersed MIP1 particles for ATP was carried out according to our previous work [42]. Firstly, atropine sulfate (69.5 mg, 0.1 mmol) and MAA-β-CD (541 mg, 0.4 mmol) were dissolved in 30 mL DMAC/H_2_O (20/10, *v/v*) by ultrasonic vibration for 30 min. TRIM (638 µL, 2 mmol) and AIBN (30 mg) were then added into the mixture. The flask was sealed and maintained under mild stirring water bath adjusted at 60 °C for 24 h. Subsequently, the polymer particles were eluted with methanol and centrifuged at 13,000 rpm for 10 min. Finally, the template was removed with methanol/acetic acid (80/20, *v/v*) using Soxhlet extraction until no template molecules could be detected. The residual acetic acid was removed from the MIPs by washing with methanol and ultrapure water. 

MIP2, and MIP3 were prepared following the same procedure described above; however, the cross-linker was replaced with EPI (156.41 µL, 2 mmol, [43]) and genipin (452.46 mg, 2 mmol, [44]), respectively. The corresponding non-molecularly imprinted polymers (NIPs) were prepared via the same synthetic route, but in the absence of template molecules. 

### 2.5. Polymers Characterization

The successfully synthesized NIPs and MIPs were confirmed using FT-IR spectrophotometer (Philips Analytical, Cambridge, UK) over the range 400–4000 cm^−1^; all polymers were compressed into KBr pellets. The morphologies of the synthesized spheres were measured by field emission scanning electron microscope (SEM) (Hitachi S-4800, Hitachi Limited, Tokyo, Japan) operated at 5 kV. The samples were coated with a thin gold film before analysis.

### 2.6. Water Swelling Behavior 

The swelling studies of the prepared microspheres were carried out as follows. The water binding capacity of MIP0/NIP0 and MIP1/NIP1 was assessed in 0.9% NaCl solution at 37 °C. Microspheres (5 mg) were dried for one day and the sorption behavior was monitored by detecting the increase in mass of the samples at different time intervals. After wiping liquid off from the surface of the MIP beads with Kimwipe paper (Kimberly Clark Professional, Roswell, GA, USA), the swelling ratio (SR) was calculated as follows: (1)$$\mathrm{SR}\;\left(\%\right)=\frac{W_{t}-W_{o}}{W_{o}}\times 100\%,$$
where *W*_o_ (mg) and *W*_t_ (mg) are the weights of the original and swollen samples at time t, respectively. 

### 2.7. Drug Loading Experiment

To assess the adsorption performance of the MIPs, static adsorption experiments were conducted over specified concentration range of ATP aqueous solution. A 2.5 mg of MIPs and various concentrations of ATP (100, 200, 400, 600, 800, and 1000 ng·mL^−1^) were vigorously shaken for 2 h using a mechanical shaker. At the end of the equilibrium time, the tubes were centrifuged and the supernatant was analyzed using LC-MS/MS. The amount of the template rebound for each polymer (*Q*, ng·mg^−1^) was calculated using Equation (2) [45]: (2)$$Q=\left(C_{0}-C_{e}\right)V/m,$$
where *C*_0_ (ng·mL^−1^) and *C*_e_ (ng·mL^−1^) are the initial and final concentrations of the incubated solution, respectively, *m* (mg) is the mass of the polymer added, *Q* (ng·mg^−1^) is the amount of the template rebound for each polymer, and *V* (mL) is the volume of the solution. All experiments were conducted in triplicate.

Furthermore, imprinting efficiency (*I*e) was adopted to evaluate the selectivity of MIP and NIP. This was obtained from the ratio of *Q*_MIP_ to *Q*_NIP_ using the following Equation (3):(3)$$I_{e}=Q_{\mathrm{MIP}}/Q_{\mathrm{NIP}},$$
where *Q*_MIP_ and *Q*_NIP_ represent the respective adsorption quantities for MIPs and NIPs. 

The selective recognition ability of MIP for ATP and its analogue anisodine (ASD) and non-analogue rutin (RU) was evaluated using the same procedure aforementioned. 

To investigate the adsorption equilibrium of ATP in MIPs, Langmuir and Freundlich isotherms were used. The linear Langmuir [46] and Freundlich [47] adsorption isotherms were calculated based upon Equations (4) and (5), respectively.
(4)$$\frac{C_{e}}{Q_{e}}=\frac{1}{bQ_{m}}+\frac{C_{e}}{Q_{m}},$$
(5)$$\mathrm{In}Q_{e}=\mathrm{In}K_{f}+\frac{1}{n}\left(\mathrm{In}C_{e}\right),$$
where *C_e_* (ng·mL^−1^) is final concentration of the incubated solution, *Q_e_* is the adsorption amount (ng·mg^−1^) at equilibrium, *Q_m_* is the adsorption volume of saturation, *b* represents enthalpy of sorption (change with temperature). *K*_f_ and n are the Freundlich constants related to the adsorption capacity and intensity, respectively. 

### 2.8. In Vitro Drug Release Test

ATP-controlled release profile of MIP in vitro was conducted by adding 5 mg of ATP-loaded MIP microspheres in simulated gastric acid of pH 1.5 and intestinal fluid of pH 7.4. The artificial liquids were prepared according to previous study [48]. The tests were conducted in 10 mL solution using mechanical shaker, at 180 rpm and 37 °C maintained condition. At specified time intervals (0.5–68 h), samples were centrifuged and re-dispersed into another 1 mL fresh simulated gastric or intestinal fluid to continue the experiment. Following each time interval, supernatant from each sample was collected for determination of ATP concentration using LC-MS/MS. NIP spheres as control groups were evaluated in the same way. The cumulative release (CR) was calculated according to Equation (6):(6)$$\mathrm{CR}\;\left(\%\right)=\frac{10.0C_{n}+1.0\sum C_{\left(n-1\right)}}{W_{o}}\times 100\%,$$
where *W*_o_ (mg) is the weight of ATP in the polymer, *C*_n_ (ng·mL^−1^) and *C*_(n−1)_ (ng·mL^−1^) are ATP concentration in solution collected at n and n−1 times, respectively. The constant 10.0 represents total detected volume, and 1.0 stands for withdrawn and replaced fresh solution.

## 3. Results and Discussion

### 3.1. Characterization and Morphology of the Polymeric Microspheres

The FT-IR spectra of the MIP and NIP microspheres (without ATP) and MIP loaded with ATP were analyzed and are shown in Figure 2. There were no differences between MIP and NIP spheres. On the other hand, we clearly defined the difference between MIP spheres with and without ATP. The FT-IR spectra of MIP sphere (with ATP) revealed the presence of –OH at 2966 cm^−1^, and skeletal vibration of phenyl at 1636 cm^−1^ and 1468 cm^−1^ and substitutional phenyl at 900–650 cm^−1^; thereby we confirm that ATP was successfully bonded with polymers. The vanishing of the C=C at 1616 cm^−1^ shows that the crosslinking response occurred between TRIM cross-linker and MAA-β-CD [49]. The characteristic peaks of MIPs (in the FT-IR spectrogram) compared with that for NIPs successfully confirmed the pre-polymerization between the MAA-β-CD monomer and the template molecules. 

We further used the scanning electron microscope to elaborate the surface structure of MIP and its equivalent NIP microspheres. Figure 3 depicts the effect of initial monomers and cross linkers on the uniformity and morphology of microspheres. It is well known that the structure of MIPs and NIPs was strongly influenced by solvents and functional monomers [50]. The structure of various polymers displayed in Figure 3 speculated that the β-CD moiety could be lodged inside the whole polymer matrix composite (PMC) of MIP1 and NIP1, which could influence the particle growth and morphological analysis [50]. When the initial monomer was shifted from MAA to MAA-β-CD, the pore diameters of MIP and NIP spheres were increased and became more uniform in shape, with the size of 600–800 nm. Introducing MAA-β-CD expanded the binding site. Thus, the chance of self-aggregation could increase by increasing the number of atoms in reaction medium. There was no obvious spherical shape when EPI and genipin were used as cross linkers. Additionally, MIP2 gave the lowest yield of 12% (Table 2). This phenomenon might be ascribed to the strong electron-withdrawing property of chlorine group from EPI that would make the interaction with pre-polymers during crosslinking process inherently difficult. 

### 3.2. Swelling Kinetics

Dried polymers retained large amounts of 0.9% NaCl solution through the first 4 h of the assay and swelling equilibrium was achieved by 24 h. Two monomers (MAA and MAA-β-CD) were used to generate molecularly imprinted polymers, and the swelling profiles of the MIPs and NIPs in water are compiled in Table 2. The MIPs exhibited higher swelling capacity than the NIPs. This behavior might be attributed to the higher porosity of the imprinted polymers [51]. The amounts of ATP loaded onto microspheres prepared without β-CD (MIP0/NIP0) was smaller than MIP1/NIP1. This is not only due to the low binding affinity of the polymers (to the template molecules) but also due to the low water retainability (Figure 4). The highest degree of swelling of MIP1/NIP1 makes the template molecules easily accessible within the polymer, efficiently anchored by the β-CD cavities, and finally loading into the microspheres [52]. The swelling value of polymers prepared with β-CD was found to be 1154%, which clearly indicates that all MAA-β-CD monomers were implemented successfully to form cross-linked polymers, exhibiting excellent superiority in swelling ability. It can be demonstrated in a similar way that the swelling of both MIP0 and MIP1 surpassed that of non-imprinted ones, owing to cavities from the template.

### 3.3. Binding Behavior of MIPs Prepared from Different Monomers

To gain an insight into the accessibility and preponderance of MAA-β-CD over MAA for MIPs and corresponding NIPs, the ability of two different series of polymers in binding ATP was tested in aqueous medium. Figure 5 illustrates that MAA (as a basic monomer) lowered the binding affinity of MIP0 to ATP, whereas MAA-β-CD (functionalized monomer) increased the drug binding tendency to MIP1. Similarly, the adsorption capacity of MIP1/MIP0 was superior to NIP1/NIP0. Negative-charged carboxylic acid group (from MAA) within the polymer matrix imparted a high affinity of polymer toward hydroxyl groups in ATP structure. Furthermore, the outstanding binding ability also benefitted from hydrophobic cavities in β-CD to form selectively complexes with various drugs by host-guest interactions. These results prove that the imprinted cavities substantially increased the binding affinity of hydrogels toward drug in aqueous media.

### 3.4. The Loading Capacity of MIP and NIP Microspheres

The amount of ATP bound to the polymer was calculated as ng of ATP mg^-1^ dry microspheres. The loading capacity of MIP was correlated to the initial ATP concentrations; the adsorption capacity increased with increasing the initial drug concentrations [53,54]. To investigate the adsorption equilibrium of ATP in MIPs, Langmuir and Freundlich isotherms were used. To determine the best fit isotherm model, the correlation coefficient (*R*^2^) for each parameter were used to evaluate the data. The corresponding parameters calculated from the above two models are listed in Table 3. As Figure 6 shown, the calculated correlation coefficients for Langmuir model on MIP1 was not satisfactory enough (*R*^2^ = 0.91). To overcome the problem regarding the best equilibrium model for MIPs [55], a Freundlich model was selected to re-fit the experimental results [56]. Indeed, the K_f_ constant calculated after fitting gave the superiority of MIP1 when compared with MIP2 and MIP3 (data not shown). Moreover, the calculated correlation coefficients for this model were much higher (*R*^2^ = 0.99). 

### 3.5. Selectivity Study

The slopes of the Scatchard plots for each isotherm were calculated using Equation (7):(7)$$\frac{Q}{C_{e}}=\frac{Q_{max}-Q}{K_{d}},$$
where *Q_max_* is the ostensible extreme amount of binding sites and K*_d_* is the dissociation constant.

The structures of the analytes used in selectivity study are shown in Figure 7a. Figure 7b clearly indicates that the adsorption ability of MIP1 was higher than NIP1. It also shows that the adsorption capacity of the polymers was enhanced by increasing the concentration of ATP. Additionally, it shows MIP1 and NIP1 with *I*_e_ values of 3.55 and 1.45 for ATP and ASD, respectively. For the comparison of selectivity of MIP1 to non-analogue and tropane alkaloids, rutin was used for confirmation with carrying out (*I*_e_) value of 1.10. The sensitivity of MIP1 for different analytes was in the order of ATP > ASD > RU. Results from Figure 7 show that the prepared MIP1 has a favorable specificity to ATP template. The value of ASD was slightly lower, while microspheres showed barely improvement in selectivity for rutin. To evaluate the reuse ability, the above procedure with the same conditions (firstly adsorption and then desorption) was repeated five times. In sum, MIP displayed good selectivity for template molecule than its analog and non-analog, and its selectivity was much better than the NIPs.

### 3.6. The Behavior of In Vitro Drug Release at Different pH 

As stated above, MIP microspheres have the capacity to give extra template molecules in comparison with NIP; accordingly, the profiles of MIP1/NIP1 were further evaluated to test whether ATP release could also be affected by pH. The release profiles of microspheres loaded with ATP in virtual gastric (pH 1.5) and intestinal liquid (pH 7.4) within the first 68 h were collected and analyzed. Under the two testing conditions of both simulated gastric and intestinal fluid, the ATP release rate of NIP1 was a little higher than MIP1 at certain time intervals (Figure 8). A burst release was observed for NIP1 in simulated gastric (pH 1.5) and the cumulative release percent of NIP1 in the first 3 h was 33.9% higher compared with MIP1. Homoplastically, the released phenomenon of microspheres in intestinal fluid was in line with that in simulated gastric fluid; this accounts for specific adsorption ability of molecularly imprinted polymers. There were specific and non-specific interactions between the ATP and the binding sites in MIPs. Therefore, drugs loaded in NIPs were quickly released in the first 12 h and achieved complete release by 28 h. As for MIPs, the release of ATP in simulated gastric fluid (pH 1.5) was faster (11 h for 50%) than that in the intestinal fluid. As the ATP molecules and hydroxyl group (from β-CD in prepared polymers) were ionized in simulated gastric fluid (pH 1.5), the high ionic strength can overcome the electrostatic interactions between ATP and polymers, therefore facilitating the release of ATP from MIPs. However, some ionic associations are affected by certain drug affinities to the β-cyclodextrin cavity, providing a poorly controlled release process by the molecular impression matrix in the fluid at pH 7.4.

A comparison of the developed MIP method with the previously-reported MIP methods for drug delivery and target-activated release systems is shown in Appendix A. As noted, the developed MIP exhibited a high drug-loading capacity and a good pH-responsive capacity in vitro, which demonstrates its potential application in drug delivery systems.

## 4. Conclusions

In the present study, a novel pH-sensitive ATP molecularly imprinted polymers microsphere was successfully prepared based on methacrylic acid functionalized beta-cyclodextrin as a functional monomer in aqueous media. Binding ability of MIP for ATP showed that MAA-β-CD as monomer had superiority over MAA in serving as drug delivery. In vitro loading and release experiments have shown that MIP not only has a higher ATP loading capacity, but also has a sustained activity than that of NIP. In the range 1.5–7.4 pH, the MIPs developed sensitive pH ATP release profile. Synthetic MIPs showed a substantial response at pH level, that is, the release rate of ATP was slightly higher and the release was much higher at low pH. The protocol revealed in this study suggests that this drug delivery system in which MIP is produced is very promising polymeric device for release and selection of ATP and possibly any target compound.

## Figures and Tables

**Figure 1 polymers-12-00130-f001:**
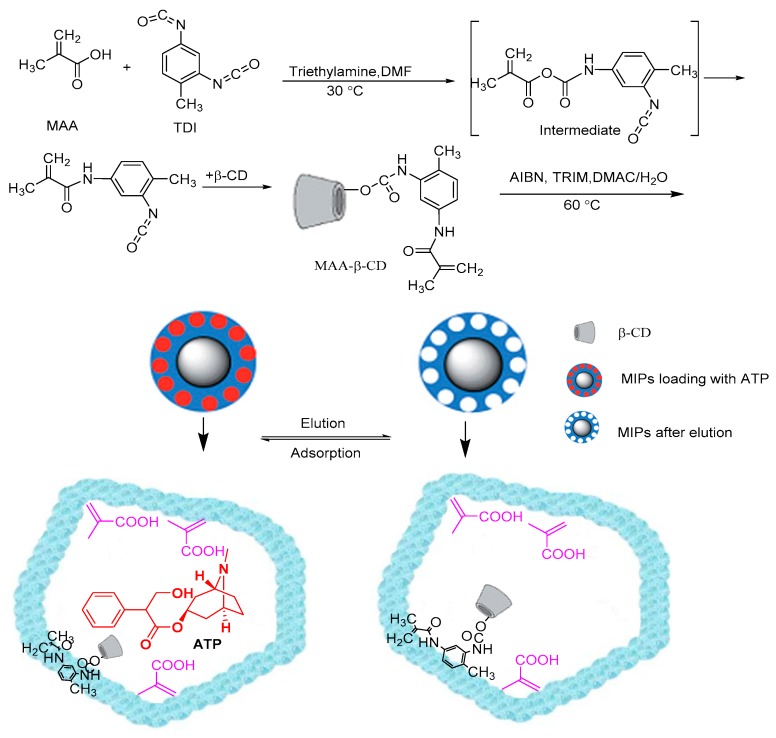
A scheme for preparation of ATP molecularly imprinted polymers (MIPs).

**Figure 2 polymers-12-00130-f002:**
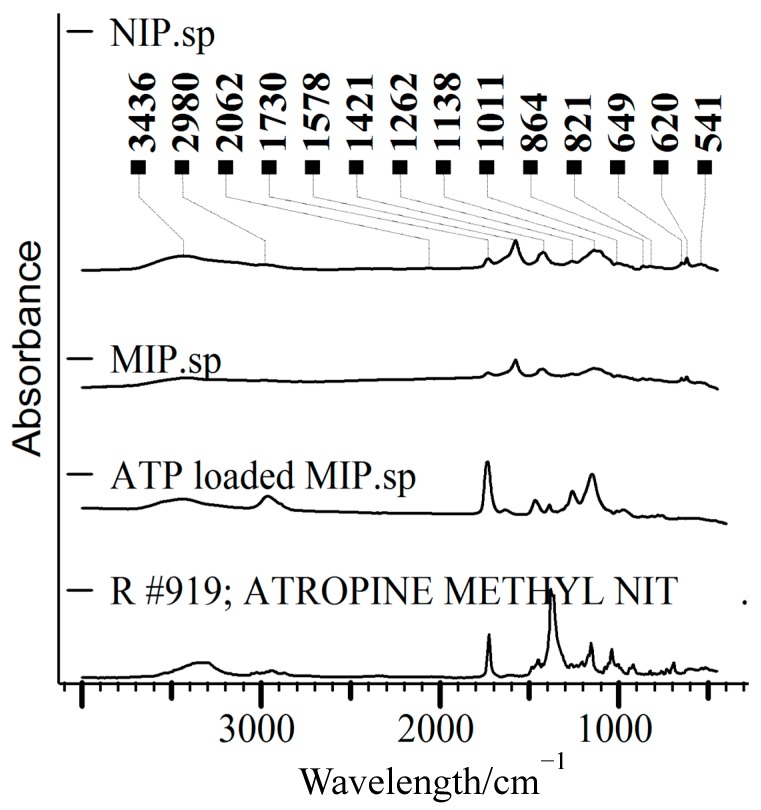
Infrared (KBr pellet) spectra of non-molecularly imprinted polymer (NIP), MIP, ATP-loaded MIP, and ATP.

**Figure 3 polymers-12-00130-f003:**
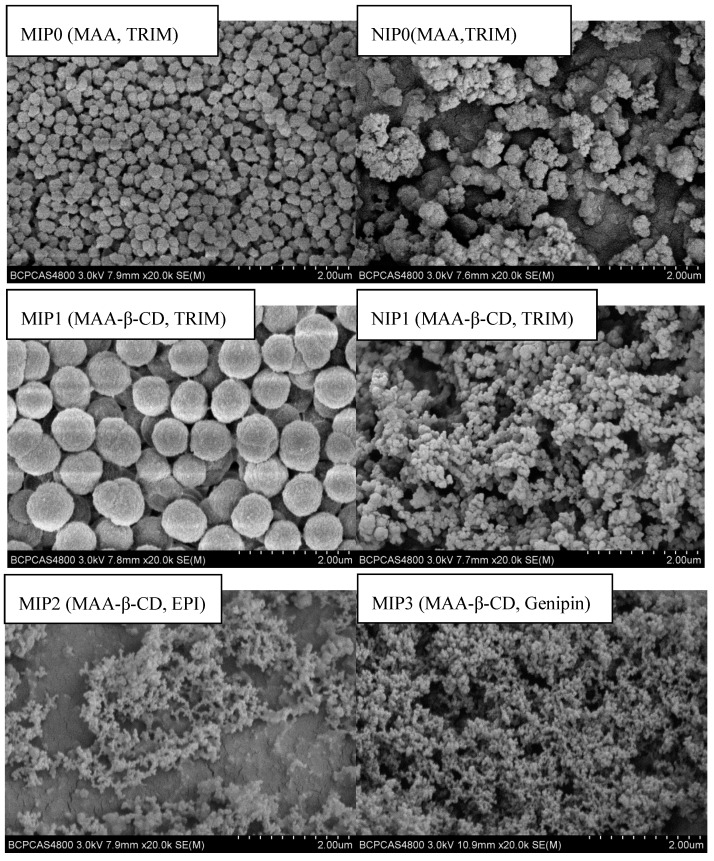
Scanning electron micrographs of different microspheres.

**Figure 4 polymers-12-00130-f004:**
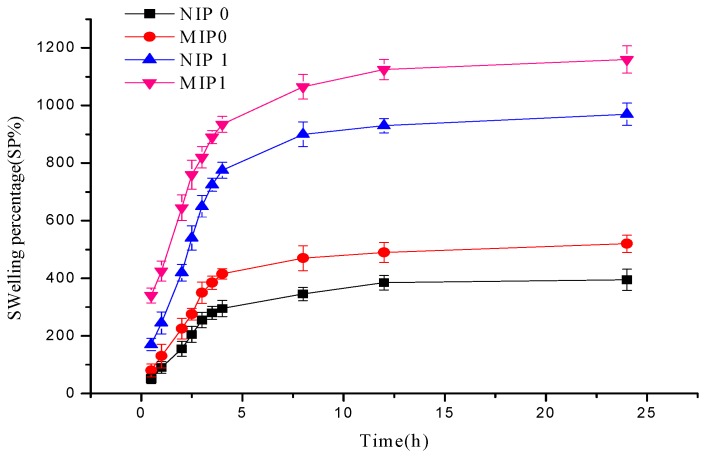
Swelling profiles of MIP0/NIP0 (MAA as monomer) and MIP1/NIP1 (MAA-β-CD as monomer).

**Figure 5 polymers-12-00130-f005:**
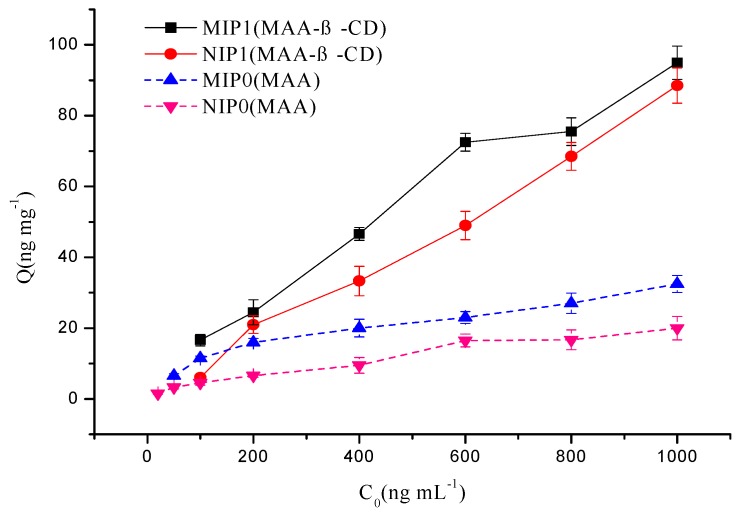
ATP binding to MIPs and NIPs with different monomers (MAA and MAA-β-CD) in aqueous media. Each data represents mean ± SEM (*n* = 3).

**Figure 6 polymers-12-00130-f006:**
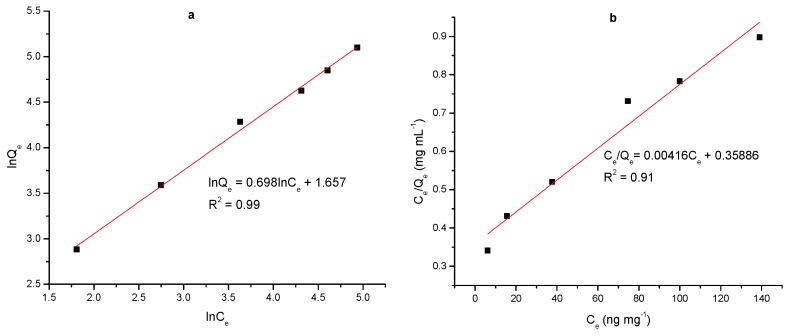
Freundlich model (**a**) and Langmuir model (**b**) MIP1.

**Figure 7 polymers-12-00130-f007:**
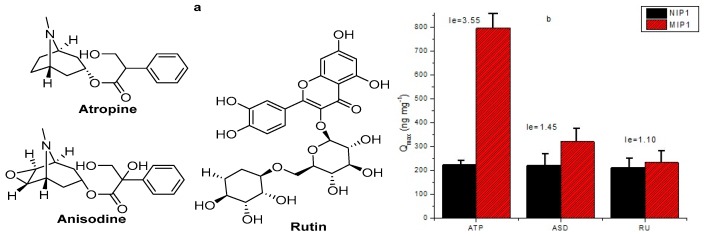
(**a**) Chemical structures of the analytes used in the selectivity study (**b**) The specificity adsorption of ASD-MIP and ASD-NIP.

**Figure 8 polymers-12-00130-f008:**
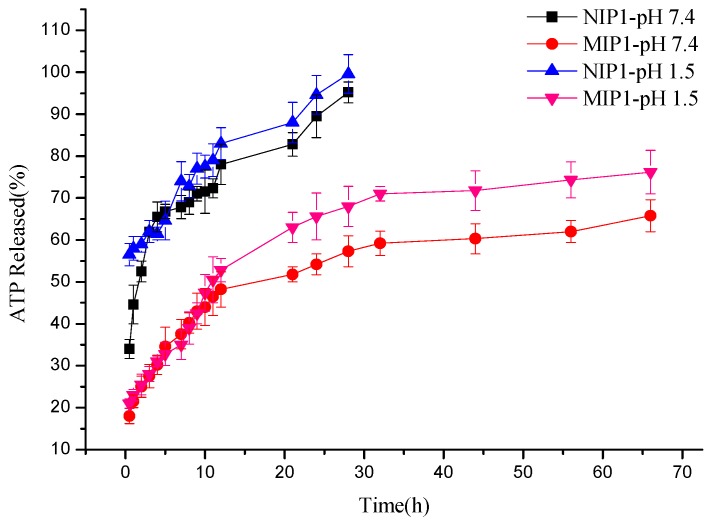
ATP released profiles in intestinal fluid (pH 7.4) and simulated gastric fluid (pH 1.5).

**Table 1 polymers-12-00130-t001:** Tandem mass spectrometry (MS/MS) parameters for determination of atropine sulfate (ATP) and anisodine (ASD).

Compounds	Parent mass (*m/z*)	Daughter mass (*m/z*)	Declustering potential (V)	Collision energy (eV)
ATP	290.2	124.1 *	69	36
		93.0		47
ASD	320.0	156.0 *	69	27
		138.1		33
		119.1		35

***** Quantitative ion.

**Table 2 polymers-12-00130-t002:** Composition of the polymers.

Microspheres name	Template (ATP)	Functional monomer	Cross-linker	ATP: monomer molar ratio	Yield (%)	Saturated swelling (%)
NIP0	-	MAA (0.4 mmol)	TRIM (2 mmol)	-	74	394
MIP0	0.1 mmol	MAA (0.4 mmol)	TRIM (2 mmol)	1/4	71	502
NIP1	-	MAA-β-CD (0.4 mmol)	TRIM (2 mmol)	-	72	983
MIP1	0.1 mmol	MAA-β-CD (0.4 mmol)	TRIM (2 mmol)	1/4	74	1154
MIP2	0.1 mmol	MAA-β-CD (0.4 mmol)	EPI (2 mmol)	1/4	12	-
MIP3	0.1 mmol	MAA-β-CD (0.4 mmol)	Genipin (2 mmol)	1/4	50	-

**Table 3 polymers-12-00130-t003:** Parameters Freundlich and Langmuir model of MIP1.

Langmuir model		Freundlich model	
*Q*_m_ (ng·mg^−^^1^)	240.8	K_f_	5.24
*b*	0.012	*n*	1.42
*R* ^2^	0.91	*R* ^2^	0.99

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
