# Peer review of "Development of Water-Compatible Molecularly Imprinted Polymers Based on Functionalized β-Cyclodextrin for Controlled Release of Atropine"

_polymers, 2020, doi:10.3390/polym12010130_

Round 1

Reviewer 1 Report

The manuscript submitted by Yahui Hea et al. describes the preparation of atropine imprinted polymers for drug controlled release. The authors developed several synthetic approaches (Table 2) however further investigated mainly polymers 0 and 1 and it should be explained why polymers 2 and 3 were not investigated. I also think that pore size and pore volume of the fabricated polymers should be investigated in order to clarify if the observed differentiation between MIP and NIP is due to imprinting effect (formation of specific cavities and specific affinity to template expressed by Kd) and not due to the observed difference in polymer morphology.        

Moreover:    

The introduction can be improved by more information on MIP applications in controlled drug release. The recovery of the template from imprinted polymer by washing should be estimated. The swelling ratio should be determined by the ratio of swollen polymer volume to the volume of dry polymer and not by the polymer weight ratio before and after solvent absorption.

Minor remarks

Page 2, lines 88-89  “In this study, Methacrylic acid functionalized β-cyclodextrin as a novel monomer was synthesized” – the synthesis of the methacrylic acid functionalized β-cyclodextrine has been previously described in 1996 by Sreenivasan as mentioned by the authors.

Page 4, lines 140-141 “Finally, the dummy template was removed with methanol/acetic acid (80/20, v/v) using Soxhlet extraction” – why dummy? You used as template the drug of interest – atropine.

Page 10, line 291 “and Kd (μM) is the ostensible separation constant” - Kd in Scatchard equation is dissociation constant.

Author Response

Response to Reviewer 1 Comments

Comments and Suggestions for Authors

Point 1:

The manuscript submitted by Yahui Hea et al. describes the preparation of atropine imprinted polymers for drug controlled release. The authors developed several synthetic approaches (Table 2) however further investigated mainly polymers 0 and 1 and it should be explained why polymers 2 and 3 were not investigated. I also think that pore size and pore volume of the fabricated polymers should be investigated in order to clarify if the observed differentiation between MIP and NIP is due to imprinting effect (formation of specific cavities and specific affinity to template expressed by Kd) and not due to the observed difference in polymer morphology.        

Response 1:  Thanks for your comment. In this study, we did prepare a series of polymers to investigate the effect of functional monomer/cross-linking monomer ratio in polymerization on the adsorption capacity. Detailed morphology of MIP and its corresponding NIP microspheres are shown in Table 2. Obviously, no spherical appearance was attained when both EPI and genipin were used as crosslinker in polymers 2 and 3. Additionally, MIP2 gave the lowest yield of 12% (Table 2). This phenomenon might be ascribed to the strong electron-withdrawing property of chlorine group in EPI that would make the interaction with pre-polymers during crosslinking process inherently difficult. Furthermore, to clarify the difference between MIP and NIP, we did the relative analysis including Langmuir, Freundlich, and nitrogen adsorption/desorption, respectively. The results showed that the K constant and imprinting factor (5.48 mL/g and 3.55) of MIP1 were superior compared to MIP2 and MIP3. On the other hand, the pore diameter and pore volume of the fabricated polymer (MIP1) obtained by nitrogen adsorption-desorption analysis were higher than that of NIP1. However, there is no significant difference among other polymers. Ongoing research undergoes to improve the instrumental conditions to further investigate the characteristics of MIP in practical application.

Point 2:

The introduction can be improved by more information on MIP applications in controlled drug release. The recovery of the template from imprinted polymer by washing should be estimated. The swelling ratio should be determined by the ratio of swollen polymer volume to the volume of dry polymer and not by the polymer weight ratio before and after solvent absorption.

Response 2:  Thanks a bunch for your suggestion. We have added new references on MIP applications in controlled drug release in the introduction section (Please see line 130-132). To thoroughly wash off the template from the imprinted polymer, various solvents such as methanol, methanol/acetic acid (90:10, v/v), and their combination were screened. It was noticed that methanol/acetic acid (80:20, v/v) would provide the highest elution efficiency than methanol and methanol/acetic acid (90:10, v/v) separately. We also tried to evaluate the swelling ratio of MIP by the volume changes of polymer; however this method gave some interference that would affect the final results. Therefore according to Zhu et al. article (Zhu, D., Chen, Z., Zhao, K., Kan, B., Li, H., Zhang, X.,et al., (2015). RSC Advances, 5(34), 26977–26984) we did estimate swelling ratio by the polymer weight ratio before and after solvent absorption.

Point 3:

Page 2, lines 88-89  “In this study, Methacrylic acid functionalized β-cyclodextrin as a novel monomer was synthesized” – the synthesis of the methacrylic acid functionalized β-cyclodextrine has been previously described in 1996 by Sreenivasan as mentioned by the authors.

Response 3: Thank you. The sentence has been amended. Please see line 95-96.

Point 4:

Page 4, lines 140-141 “Finally, the dummy template was removed with methanol/acetic acid (80/20, v/v) using Soxhlet extraction” – why dummy? You used as template the drug of interest – atropine.

Response 4: Thanks to the diligent reviewer for the good catch. The word dummy has been deleted from the sentence. Please refer to line 163.

Point 5:

Page 10, line 291 “and Kd (μM) is the ostensible separation constant” - Kd in Scatchard equation is dissociation constant.

Response 5: Apologies for this mistake. This has been amended. Please refer to line 325 

Reviewer 2 Report

This manuscript describes the preparation of a novel pH-sensitive ATP molecularly imprinted polymers microspheres for the controlled release of atropine. In general, this article lacks depth and logic, with the polymers being characterized with only a few unimportant techniques. So it is not convincing enough to be published on Polymers. The authors are suggested to consider the following major points and carefully improve the whole manuscript before the manuscript could be resubmitted. An itemized list of these points is as follows:
1. In abstract, the description of “ATP/MAA and β-CD was 1: 4 (mol/mol)” is not clear.
2. In 2.2, the authors should supply the description of standard curve for the determination of ATP by HPLC at different concentrations.
3. The characterizations of polymers are far from being enough, so other characterizations should be added. In addition, the figure format is not standard.
4. In 2.7, “Co” and “Ce” should be “C0” and “Ce”.
5. In infrared spectroscopy, the authors stated that the imprinting process occurred. Please provide additional characterization to support the conclusion. By the way, there is a problem with the infrared spectra: the figure and text do not correspond to each other, please modify them again.
Error bars for all experiments need to be added.
6. The authors are suggested to explain the reason of fitting “Freundlich isotherm model”. And nonlinear fitting is recommended.
7. Comparison between this work and the previous work from other groups must be provided to show the advantage of this work.
8. The authors should add the selectivity study of MAA as functional monomer.
9. The literatures are too old, the authors are suggested to complement the latest literatures. And the format of references should be unified, please check again.

Author Response

Response to Reviewer 2 Comments

This manuscript describes the preparation of a novel pH-sensitive ATP molecularly imprinted polymers microspheres for the controlled release of atropine. In general, this article lacks depth and logic, with the polymers being characterized with only a few unimportant techniques. So it is not convincing enough to be published on Polymers. The authors are suggested to consider the following major points and carefully improve the whole manuscript before the manuscript could be resubmitted. An itemized list of these points is as follows:

Point 1:

In abstract, the description of “ATP/MAA and β-CD was 1: 4 (mol/mol)” is not clear.

Response 1: Thank you. The sentence has been amended. Please see line 23-24.

Point 2:

In 2.2, the authors should supply the description of standard curve for the determination of ATP by HPLC at different concentrations.

Response 2: Thank you. This has been added as suggested. Please refer to line 143-144.

Point 3:

The characterizations of polymers are far from being enough, so other characterizations should be added. In addition, the figure format is not standard.

Response 3: Thank you. In this study, the developed MIP was characterized by Fourier-transform infrared spectroscopy and scanning electron microscopy. Furthermore, the adsorption capacity of MIP was evaluated using adsorption kinetics, isotherms, selectivity, and recycling experiments. We also did the relative analysis including Langmuir, Freundlich, and nitrogen adsorption/desorption , respectively.

In revised manuscript, we added the figure of 1H NMR spectrum. The developed methacrylic acid functionalized β-cyclodextrin (MAA-β-CD) was confirmed by 1H NMR spectroscopy. The 1H NMR (DMSO-d6) as following Figure showed a peak on 1.98 (Ha), which is allocated to CH3 proton, and one of the peak at 5.72 and 5.79 (Hb) was attributed to the C=C protons. Peaks at 9.64 (Hc) and 9.86 (Hd) suggest two secondary amines coming from TDI linker. On the other hand, peaks at 7.32 (He), 7.36 (Hf), and 7.72 (Hg) are representing protons of benzene ring. At peak 2.19 (Hh), the tolyl set was detected for methyl proton. One of the peaks found at 4.39, 5.7, and 3.5 - 3.7 were assigned to Ha'-Hk' of tetrahydro-2H-pyran. 3.94, 4.71, 4.77, 4.77, and 4.71 were H1-H5 and assigned respectively to OH consistent to the functionalized beta-cyclodextrin (β-CD) molecular structure.

 We did revise the figures to be up to the standard.

Point 4:

In 2.7, “Co” and “Ce” should be “C0” and “Ce”.

Response 4: Thank you. This has been amended

Point 5:

In infrared spectroscopy, the authors stated that the imprinting process occurred. Please provide additional characterization to support the conclusion. By the way, there is a problem with the infrared spectra: the figure and text do not correspond to each other, please modify them again.

Error bars for all experiments need to be added.

Response 5: Thanks a bunch for raising this comment. In this study, the FT-IR spectra of the MIP and NIP microspheres (without ATP) and MIP loaded with ATP were analyzed and shown in Fig. 2. Obviously, there were no differences between MIP without ATP and NIP spheres. On the other hand, we clearly defined the difference between MIP spheres with and without ATP. The FT-IR spectra of MIP sphere (with ATP) revealed the presence of–OH at 2966 cm-1, skeletal vibration of phenyl at 1636 cm-1 and 1468 cm-1, and substitutional phenyl at 900–650 cm-1. These peaks in infrared spectroscopy contribute to the characterization of ATP, thereby confirming that ATP was successfully bonded with polymers. Furthermore, we did perform the selectivity study by the Scatchard plots of MIP. It clearly indicates that the adsorption ability of MIP1 for ATP was higher than NIP1. Generally, the characterization of MIP was mostly performed by infrared spectroscopy, please see this reference (Mao C, Xie X, Liu X, et al. The controlled drug release by pH-sensitive molecularly imprinted nanospheres for enhanced antibacterial activity[J]. Materials Science and Engineering C, 2017, 77:84-91). We will adopt additional new tools in upcoming MSs.

Error bars were added to Fig.7 as suggested.

Point 6:

The authors are suggested to explain the reason of fitting “Freundlich isotherm model”. And nonlinear fitting is recommended.

Response 6: Thank you. In our experiment, the calculated correlation coefficients for Langmuir model on MIP1 was not satisfactory enough (R2=0.91). To overcome the problem regarding the best equilibrium model for MIPs, Freundlich model was selected to re-fit the experimental results (Kyzas, et al., 2015). As reported by Rampey et.al., this model is believed to be the best equilibrium model for MIPs (A.M.Rampey, R.J.UmplebyIi, G.T.Rushton, J.C.Iseman, R.N.Shah, K.D.Shimizu, Characterization of the imprint effect and the influence of imprinting conditions on affinity, capacity, and heterogeneity in molecularly imprinted polymers using the Freundlich isotherm-affinity distribution analysis, Anal.Chem. 76 (2004) 1123–1133. This model is derived by assuming a heterogeneous surface with a non-uniform distribution of heat of adsorption over the surface. In sum, the results showed that MIPs possessed higher loading capacities.  

Point 7:

Comparison between this work and the previous work from other groups must be provided to show the advantage of this work.

Response 7: Thank you. So far, there were no published studies on the development of biodegradable MIP microspheres for in vitro controlled release of ATP. In this study, we developed β-cyclodextrin nano-sized MIP to control the release of atropine. Synthetic MIPs showed a significant response at pH level (i.e. the release rate of ATP was slightly higher and the release was much higher at low pH). The protocol revealed in this study suggests that this drug delivery system in which MIP is produced is a very promising polymeric device for release of ATP. We think that the above results should be advantageous for the present study. MIP with MAA-β-CD as monomer had superiority over MAA in serving as drug delivery compared with other MIP when MAA used as functional monomer. For example, Sina Farzaneh.et al. (RSC Adv., 2015, 5, 9154) reported that the MIPs for the controlled release of olanzapine was available in two different media (SDS 1% and PBS),which did not show a significant response at pH level.

Point 8:

The authors should add the selectivity study of MAA as functional monomer.

Response 8: Thank you. We did prepare a series of polymers to investigate the effect of amount and ratio of functional monomers (MAA and MAA-β-CD) and crosslinking in polymerization on the adsorption capacity. The MIP morphology and its corresponding NIP microspheres are detailed in Table 2. To gain an insight into the accessibility and preponderance of MAA-β-CD over MAA for MIPs and corresponding NIPs, the ability of two different series of polymers in binding ATP was tested in aqueous medium. Fig. 5 illustrates that MAA (as a basic monomer) lowered the binding affinity of MIP0 to ATP, whereas MAA-β-CD (functionalized monomer) increased the drug binding tendency to MIP1. Therefore, we did not further evaluate the selectivity of MIP0, which used MAA as a functional monomer.

Point 9:

The literatures are too old, the authors are suggested to complement the latest literatures. And the format of references should be unified, please check again.

Response 9: Thank you. The reference lists are updated and formatted according to the guide for authors.

Reviewer 3 Report

The manuscript " Development of water-compatible molecularly imprinted polymers based on functionalized β-cyclodextrin for controlled release of atropine" by Yahui He et al., the authors develop a novel molecularly imprinted polymers (MIPs) using methacrylic acid functionalized beta-cyclodextrin (MAA-β-CD) monomer as a potential water compatible composite for the controlled release of atropine (ATP). The paper fit the aims and scope of Polymers. The work is technically sound and scientifically valid. The paper seems to be acceptable but, in my opinion, it requires some modifications. Additionally, several questions should be answered by the authors in detail.

There were only two literatures commented in Introduction section were published in recent three years. It seems that the author neglects the newest reports in recent years. It is suggested that the author should pay close attention to new development. In addition, it is worth expanding the text with the inclusion complexes based on cyclodextrin and its derivative, with proper literature references including “Enhanced Solubility, Stability, and Herbicidal Activity of the Herbicide Diuron by Complex Formation with β-Cyclodextrin. Polymers 201911, 1396” and “Physicochemical properties and fungicidal activity of inclusion complexes of fungicide chlorothalonil with β-cyclodextrin and hydroxypropyl-β-cyclodextrin. Journal of Molecular Liquids, 2019, 293, 111513.”

Line 22 what is the mean of “mol%”?

Line 23 “in vitro” should be italic.

Line 75 “methacrylic acid (MAA) with β-CD (Ma, et al., 2013)..” redundant full stop should be deleted.

Line 94 TDI was first mentioned in Figure 1, which should be defined. As indicated in Figure 1, the compound bound to β-CD is not MAA, why named the product as “MAA-β-CD”. What`s more, Figure 1 shows that the inclusion complex was formed by ATP and original β-CD, instead of “MAA-β-CD”?

Line 127 Does it makes sense to control such a precise volume? The same to 567mg in Line 130. In addition, what the M in “0.5 M (567 mg)” means?

Line 161 Symbols in formulas should be accurately subscripted. Similar condition existed in almost all formulas in this manuscript.

Line 199 “there was no difference between…”

Line 215 “structure of various polymers displayed in Fig. 5 speculated that” Is any figure missed?

Line 226 The first two images in Figure 3 should be marked.

Line 261 It seems that the author just reported the means in Figure 5, not mean± SEM.

Line 268-275, Line 289-291 It should be described in “Material and methods”.

Line 303 Key experiments are recommended to be conducted for replicates and uncertainties should be reported.

Author Response

Response to Reviewer 3 Comments

The manuscript " Development of water-compatible molecularly imprinted polymers based on functionalized β-cyclodextrin for controlled release of atropine" by Yahui He et al., the authors develop a novel molecularly imprinted polymers (MIPs) using methacrylic acid functionalized beta-cyclodextrin (MAA-β-CD) monomer as a potential water compatible composite for the controlled release of atropine (ATP). The paper fit the aims and scope of Polymers. The work is technically sound and scientifically valid. The paper seems to be acceptable but, in my opinion, it requires some modifications. Additionally, several questions should be answered by the authors in detail.

Point 1:

There were only two literatures commented in Introduction section were published in recent three years. It seems that the author neglects the newest reports in recent years. It is suggested that the author should pay close attention to new development. In addition, it is worth expanding the text with the inclusion complexes based on cyclodextrin and its derivative, with proper literature references including “Enhanced Solubility, Stability, and Herbicidal Activity of the Herbicide Diuron by Complex Formation with β-Cyclodextrin. Polymers 2019, 11, 1396” and “Physicochemical properties and fungicidal activity of inclusion complexes of fungicide chlorothalonil with β-cyclodextrin and hydroxypropyl-β-cyclodextrin. Journal of Molecular Liquids, 2019, 293, 111513.”

Response 1: Thank you. Both references were added to the text as well as the reference lists. Please see Ref, 42 and 43.

Point 2: Line 22 what is the mean of “mol%”?

Response 2: Thank you. mol% is the mole ratio of TRIM in the total moles of polymerization solution. To make it clear, we did revise the sentence. Please see line 24.

Point 3:Line 23 “in vitro” should be italic.

Response 3: Thank you. The word in vitro has been italized throughout the text

Point 4 :Line 75 “methacrylic acid (MAA) with β-CD (Ma, et al., 2013)..” redundant full stop should be deleted.

Response 4: Thank you. This has been amended.

Point 5: Line 94 TDI was first mentioned in Figure 1, which should be defined. As indicated in Figure 1, the compound bound to β-CD is not MAA, why named the product as “MAA-β-CD”. What`s more, Figure 1 shows that the inclusion complex was formed by ATP and original β-CD, instead of “MAA-β-CD”?

Response 5: Thank you. We did define TDI as 2, 4-Toluene diisocyanate。and replaced β-CD by MAA-β-CD in polymer skeleton. Please see Fig. 1

Point 6: Line 127 Does it makes sense to control such a precise volume? The same to 567mg in Line 130. In addition, what the M in “0.5 M (567 mg)” means?

Response 7: Thank you. 0.5 M=567 mg by molecular weight. This has been amended

Point 7: Line 161 Symbols in formulas should be accurately subscripted. Similar condition existed in almost all formulas in this manuscript.

Response 7: Thank you. This has been amended throughout the text

Point 8: Line 199 “there was no difference between…”

Response 8:  Apologies, this sentence has been amended.

Point 9: Line 215 “structure of various polymers displayed in Fig. 5 speculated that” Is any figure missed?

Response 9: Apologies, this should be Fig. 3. This has been amended

Point 10:Line 226 The first two images in Figure 3 should be marked.

Response 9: Thank you. This has been marked as suggested.

Point 11: Line 261 It seems that the author just reported the means in Figure 5, not mean± SEM.

Response 10: Thank you. This has been amended.

Point 12: Line 268-275, Line 289-291 It should be described in “Material and methods”.

Response 11: Thank you. This has been amended as suggested

Point 13: Line 303 Key experiments are recommended to be conducted for replicates and uncertainties should be reported.

Response 12: Thank you. All experiments were conducted in triplicates, especially, the adsorption experiment of MIP1 and NIP1. We added error bars to the Figures.

Round 2

Reviewer 2 Report

In this revised manuscript, the authors have given some timely and detailed explanations or responses to the suggestions from the reviewers, which makes the manuscript more self-consistent than before. Now, only some minor works need to be done before its acceptance:
1. Error bars for all experiments need to be added.
2. The authors are suggested to further explain the reason of fitting “Freundlich isotherm model”.
3. In 3.6, please provide evidence whether the prepared polymer is harmful to the human.
4. Comparison between this work and the previous works from other groups must be provided to show the advantage of this work.

Reviewer 3 Report

The author had addressed all the comments. The manuscript was suitable for published.

Author Response

Dear  Reviewer,

Thanks a bunch for your positive and constructive comments on our revised MS entitled ″Development of water-compatible molecularly imprinted polymers based on functionalized β-cyclodextrin for controlled release of atropine" (Manuscript ID: polymers-645973). We have carefully revised the MS .The fine tune of the language has been checked carefully by one of our authorships, Pr. A. M. Abd El-Aty, who is a member of Joint FAO/WHO Expert Committee on Food Additives (JECFA). We do hope that the revised MS would be acceptable for publication in polymers.

Sincerely yours,

Prof. Yongxin She